# Light Transmission in Fog: The Influence of Wavelength on the Extinction Coefficient

**Pierre Duthon** *,†, **Michèle Colomb** † and **Frédéric Bernardin** †

Cerema, Equipe-projet STI, 8-10, rue, Bernard Palissy, CEDEX 2, F-63017 Clermont-Ferrand, France
* Correspondence: pierre.duthon@cerema.fr; Tel.: +33-4-7342-1069
† These authors contributed equally to this work.



**Featured Application: This work gives some advice on choosing the best wavelengths for active sensors working in the near-infrared spectral band (such as LiDARs or time-of-flight cameras) taking into account fog conditions.**

**Abstract:** Autonomous driving is based on innovative technologies that have to ensure that vehicles are driven safely. LiDARs are one of the reference sensors for obstacle detection. However, this technology is affected by adverse weather conditions, especially fog. Different wavelengths are investigated to meet this challenge (905 nm vs. 1550 nm). The influence of wavelength on light transmission in fog is then examined and results reported. A theoretical approach by calculating the extinction coefficient for different wavelengths is presented in comparison to measurements with a spectroradiometer in the range of 350 nm–2450 nm. The experiment took place in the French Cerema PAVIN BPplatform for intelligent vehicles, which makes it possible to reproduce controlled fogs of different density for two types of droplet size distribution. Direct spectroradiometer extinction measurements vary in the same way as the models. Finally, the wavelengths for LiDARs should not be chosen on the basis of fog conditions: there is a small difference (<10%) between the extinction coefficients at 905 nm and 1550 nm for the same emitted power in fog.

**Keywords:** fog; LiDAR; wavelength; extinction coefficient; meteorological optical range; spectroradiometer; droplet size distribution

## 1. Introduction

The development of full autonomous vehicles is a challenge, and LiDARs are central to upcoming developments. Adverse weather such as fog affects the range of target detection by the sensor. Both experimental and simulation approaches are needed to determine the performance of the sensors. Measurements need to be made in controlled conditions at various fog densities. A comparative study of various LiDAR technologies with various wavelengths (905 nm and 1550 nm) was undertaken [1]. The results did not show particularly large differences in the target detection distances between wavelengths. In order to better understand these initial results and propose new options, we examined the influence of wavelength on attenuation in fog by means of a theoretical approach (using two models: Mie and MODTRAN) and experimental validation (direct measurement with a spectroradiometer). Our study takes into account:

- the real droplet size distribution (DSD) of fog;
- two fog types;
- the meteorological optical range (MOR), denoted by $V$, from 15 m to 175 m;
- and wavelengths from 350 nm to 2450 nm.

Scientific studies concerning the extinction coefficient of light in fog have already been carried out, without fully addressing the problem: either because the wavelength ranges are limited, or because the DSD is not taken into account, or because the comparison between models and experimental measurements is not addressed. An early piece of research work concerning the study of the extinction coefficient for different wavelengths was carried out for aeronautics [2]. The objective was to have a measurement of the extinction coefficient for the visible, near-infrared, and far-infrared domains (to detect people in the fog). This initial work [2] took the DSD and Mie theory into consideration. However, since these measurements were made on natural fogs, few data were collected: only 600 measurements over eight wavelengths between 350 and 10,000 nm were obtained, compared to our 4456 measurements over 213 wavelengths between 360 and 2490 nm. In addition, most of the observations made were in hazy conditions (very light fog) and not in heavy fog conditions. The conclusions of this work are mixed: some fogs have constant extinction for wavelengths between 350 and 1700 nm, while others are said to have lower extinction for higher wavelengths. It was therefore observed that, beyond the visible range, fogs with the same MOR can have different extinction coefficients for different wavelengths. In addition, it was also observed that the extinction coefficient depends on the wavelength.

The main application of the research work on spectral light transmission through fog was free space optics (FSO). Much work has been done to compare extinction coefficients according to wavelength in the near-infrared domain. A literature review on the extinction coefficient relationship with MOR lists nine models [3]. Some of these models are purely theoretical, while others come from real measurements such as those of our study. However, all the calculations were carried out on the 850-nm wavelength only.

Other research work was carried out in the visible and near-infrared domains using different simulation software for the spectral band from 690 nm–1550 nm [4]. The purpose of that work was to determine the monochromatic radiation least attenuated by fog among the wavelengths 690 nm, 750 nm, 850 nm, and 1550 nm. According to that study, the most useful wavelength in fog conditions is 690 nm. However, this study was based solely on modeling, and there was no experimental confirmation of the results presented. In addition, the spectral band examined by the model was much narrower than that processed in our study.

The extinction of light radiation in fog and rain was also studied with the objective of seeking a relationship between MOR and signal attenuation by taking into account multiple scattering [5]. In both cases, it was found that the calculated extinction coefficient under the multiple scattering assumption (MODTRAN case) was lower than under the single scattering assumption (Mie theory). This difference was much greater for rain: the attenuation for multiple scattering was then half as great. In dense fog conditions, the extinction difference obtained by calculations based on these two assumptions was much smaller: about 1.5-times higher in the worst cases ($V < 100$ m) and equal in the best cases ($V > 300$ m). This study was done at specific wavelengths (550 nm, 850 nm, and 1550 nm) and not over an extended wavelength range, which justifies our work.

Measurements similar to the ones we propose have already been made in the laboratory and compared to existing models [6]. However, these models do not take into account the DSD of the fog produced since they are of the form $V = f(\lambda)$ where $V$ is the MOR and $\lambda$ the wavelength. As already stated, there exist different types of fog with small or large droplets, with the same MOR. It is therefore very difficult to know if the results obtained are reproducible with natural fog. In addition, the wavelength range studied was limited to 600 nm over 1750 nm. These measurements were preceded by other work, specifically at wavelengths of 830 nm and 1550 nm [7]. Those studies concluded by showing the importance of taking DSD into account in order to evaluate the exact extinction coefficient.

The most recent research work focused on LiDAR technologies. The objective was to assess whether certain near-infrared wavelengths may be more relevant in fog. In [1], a comparison between two LiDARs of different wavelengths (905 nm and 1550 nm) in fog conditions was done. This previous study concluded that both wavelengths had the same behavior in fog conditions. However, this

conclusion was deduced from a ratio calculation, because the two LiDARs did not use the same energy level at all. The one at 905 nm had an energy level 20-times lower than the LiDAR at 1550 nm for eye safety reasons. Our more complete approach will provide a direct measurement of the extinction coefficient under fog conditions for much wider wavelength ranges.

To conclude, existing research work compared the extinction coefficient in fog for isolated wavelengths only. To our knowledge, no work has taken into account all wavelengths continuously. In addition, most of the research encountered has not taken into account the DSD, which nevertheless has a very strong impact on the extinction coefficient as described by the Mie theory. Finally, existing work focused either on the modeling or the experimental aspect. Here, we take all these aspects into account, in order to produce a more complete study.

First of all, it is necessary to give some definitions about fog itself, at a macroscopic scale, by the reduction of visibility in the atmosphere (MOR), and also at a microscopic scale, by DSD. The interaction of light with fog is discussed in Section 2 considering two models for the wavelength range of 350 nm–2450 nm. The two models used are the Mie model [8] and the MODTRAN software model (which includes Mie diffusion) [9].The method used to study the influence of wavelength on light transmission in fog is given in Section 3. This method is based on a theoretical approach, based on DSD measurement, for calculating the extinction coefficient for different wavelengths, in comparison to measurements with a spectroradiometer in the range of 350 nm–2450 nm obtained in controlled fog conditions. The results are given in Section 4 and discussed in Section 5.

## 2. Theory and Definitions: Light and Fog

### 2.1. Fog Definition

According to the definition given by the World Meteorological Organization (WMO) [10], fog is the suspension in the atmosphere of microscopic water droplets that reduce MOR on the ground to less than one kilometer.

MOR is the distance through fog for which the luminous flux of a collimated light beam is reduced to 5% of its original value.

According to this definition, MOR (denoted as $V$ and expressed in meters) is related to the extinction coefficient ($k$) [10]:

$$V = -\frac{ln(0.05)}{k} \simeq \frac{3}{k} \tag{1}$$

where $k$ denotes the extinction coefficient in the visible band, more precisely defined in Section 2.3. In the general case, the extinction coefficient $k$ depends on the wavelength. In the visible domain, $k(\lambda)$ is considered as constant by the WMO [10]. This is not the case for infrared wavelengths [2].

The lower the MOR, the denser the fog and the higher the extinction coefficient.

### 2.2. Different Kinds of Fog

In nature, the two most representative kinds of fog are defined according to the formation mechanism:

- Radiation fog forms overnight or early in the morning. This type of fog generally dissipates in the morning. It frequently flows into valleys or low-lying areas. It may then also be qualified as "continental fog". Radiation fog is composed of small droplets, with diameters distributed around a mode between one and 10 microns.
- Advection fog forms when a moist warm air mass is pushed over a relatively cold surface. This type of fog can last all day. Advection fog may form over oceans and move to continents where the ground is colder. It may then also be qualified as "maritime fog". Advection fog is composed of large droplets, with diameters distributed around a mode between 10 and 20 microns.

The DSD, denoted as $n(r)$, characterizes the micro-structure of fog [11–13]. It gives the number of particles per unit of volume $n(r)\,dr$, generally expressed in cm$^{-3}$, whose radius is in the radius

class between $r$ and $r + dr$, where $dr$ is the radius width of the class, generally expressed in µm. $n(r)$ is generally expressed in $\text{cm}^{-3} \cdot \text{µm}^{-1}$. The DSD can be measured by a particle size analyzer (PSA), which is an optical particle counter.

Figure 1 shows the models of the two most representative kinds of dense fog corresponding to heavy advection fog (Model 1) and heavy radiation fog (Model 3) of [11]. Models 2 and 4 of [11] were not used here because they correspond to moderate fogs. It is important to note that the measuring devices on which these models were calibrated (in 1979) could not detect the smallest drops (less than 1 µm). These models therefore tended to underestimate the total number of drops, but also to overestimate the mean size of the drops. A land vehicle, particularly an autonomous vehicle, is likely to encounter these two types of natural fog if it moves within continental and seaside areas.

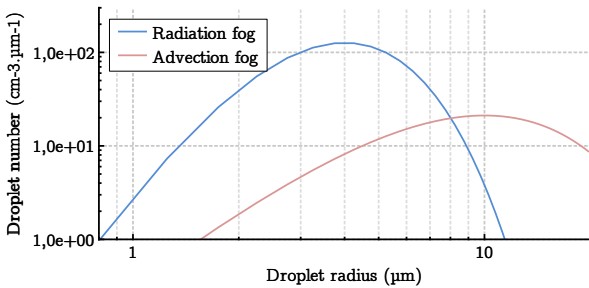

**Figure 1.** Droplet size distribution models for radiation and advection fogs with $V = 50$ m [11].

*2.3. Light Attenuation Due to Fog*

In foggy conditions, a luminous flux of wavelength $\lambda$ is attenuated according to the Bouguer–Lambert law [10]:

$$\Phi = \Phi_0 . e^{-k(\lambda).d} \tag{2}$$

where $k(\lambda)$ is the extinction coefficient and $d$ is the distance between the source (emitting the luminous flux $\Phi_0$) and the observer (receiving the luminous flux $\Phi$).

This attenuation is due to the interaction of light radiation with microscopic water droplets [14,15]. It is divided into two complementary phenomena: absorption and scattering.

The absorption phenomenon is caused by the interaction between an electromagnetic wave and the electronic cloud of an atom. This phenomenon has the consequence of absorbing part of the incident energy. This part of the energy is conserved by the atom in interaction with the radiation.

The scattering phenomenon is caused by the change in the trajectory of the luminous flux caused by fog microscopic water droplets. Scattering can be modeled using Mie theory [8]. This theory is a particular solution of Maxwell equations describing the elastic scattering of a plane electromagnetic wave by a spherical particle. This theory applies with the following assumptions:

- a monochromatic incident light;
- a spherical, homogeneous, isotropic particle of radius $r$, of refractive index $n_1$;
- non-absorbing dispersion host environment of refractive index $n_2$;
- a low concentration (single scattering). This assumption will be discussed further, compared with the MODTRAN model, which includes multiple scattering.

The particle studied is a sphere of radius $r$ and refractive index $n_1$ (see Figure 2). This particle is placed in a host environment of refractive index $n_2$ in which a monochromatic plane wave with wavelength $\lambda$ propagates with a wave vector $\vec{\ell}$ such that $\|\vec{\ell}\| = 2\pi/\lambda$. This incident wave meets the spherical particle, and the radiation is scattered in all directions as a result of this interaction. The polarization of the wave is identified by the scattering plane. The latter is defined by the direction of the wave vector $\vec{\ell}$, as well as the observation direction $\vec{OP}$.

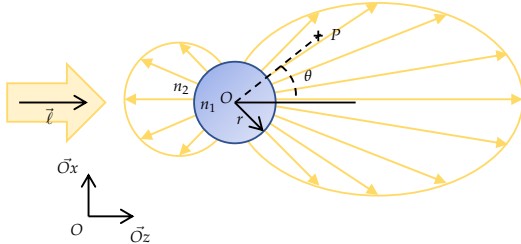

**Figure 2.** Geometric representation of the diffusion from a spherical particle. In yellow, a schematic representation of the Mie diffusion by a spherical particle.

The size parameter $x$ determines the diffusion and is defined by [8]:

$$x = \|\vec{\ell}\|\, r = \frac{2\pi r}{\lambda} \tag{3}$$

The angular distribution of the scattered radiation of spherical particles depends on their size. In the case of particles with a radius of $r << \lambda$, Mie theory and Rayleigh scattering give the same results. Rayleigh scattering is valid only for particles with a radius $r$ less than $\lambda/10$. Mie theory is valid for all the values of radius $r$ provided that the particle is spherical. In the following, Mie theory will be applied to fog droplets in the Mie + PSA model, whereas the MODTRAN + PSA model will include Mie theory applied to fog droplets and Rayleigh theory to molecules. The amount of energy scattered forward is greater than in any other direction according to Mie theory, as shown in Figure 2.

The extinction phenomenon is the combination of absorption and scattering. In order to study the extinction of radiation, it is necessary to calculate the extinction efficiency factor $Q_{ext}$. The calculation of $Q_{ext}$ requires the determination of the complex Lorenz–Mie coefficients $a_n$ and $b_n$ [16]:

$$a_n = \frac{\gamma^2 j_n(\gamma x)\,[x j_n(x)]' - j_n(x)\,[\gamma x j_n(\gamma x)]'}{\gamma^2 j_n(\gamma x)\,\left[x h_n^{(1)}(x)\right]' - h_n^{(1)}(x)\,[\gamma x j_n(\gamma x)]'} \tag{4}$$

$$b_n = \frac{j_n(\gamma x)\,[x j_n(x)]' - j_n(x)\,[\gamma x j_n(\gamma x)]'}{j_n(\gamma x)\,\left[x h_n^{(1)}(x)\right]' - h_n^{(1)}(x)\,[\gamma x j_n(\gamma x)]'} \tag{5}$$

where $x$ is the size parameter, $\gamma = n_1/n_2$ is the index ratio, $h_n(z) = j_n(z) + i y_n(z)$, and $j_n(z)$ (resp. $y_n(z)$) is the Bessel function of the first kind (resp. second kind) of order $n$ and argument $z$. The coefficients $a_n$ and $b_n$ are then used to calculate the extinction efficiency factor $Q_{ext}$:

$$Q_{ext} = \frac{2}{x^2} \sum_{n=1}^{\infty} (2n+1) Re(a_n + b_n) \tag{6}$$

where $Re(z)$ denotes the real part of the complex number $z$. In practice, this infinite series can be truncated at a rank $n_{max}$:

$$Q_{ext} = \frac{2}{x^2} \sum_{n=1}^{n_{max}} (2n+1) Re(a_n + b_n) \tag{7}$$

The proposed value for $n_{max}$ is [16]:

$$n_{max} = x + 4x^{1/3} + 2 \tag{8}$$

The extinction efficiency factor $Q_{ext}(\lambda)$ characterizes the extinction of a single particle in all directions. In order to take into account the extinction in fog resulting from the action of several particles, the extinction coefficient $k(\lambda)$ is defined from the droplet size distribution of the fog. Mie theory makes it possible to determine the expression of this coefficient. Under the hypotheses of an

unpolarized incident wave, the absence of multiple scattering of the wave on the scatterers present in the environment and independence between the scatterers (assumed to be quite distant from each other), the extinction coefficient $k(\lambda)$ is expressed in $\text{m}^{-1}$ as:

$$k(\lambda) = 10^5 \int_0^\infty Q_{ext}(x,\gamma) \pi r^2 n(r) dr \qquad (9)$$

where $\lambda$ is the wavelength in the host environment, expressed in meters, $x = 2\pi r/\lambda$ is the size parameter, $r$ is the radius of the droplet, expressed in meters, $n(r)$ is the fog DSD, expressed in particles/cm$^3$, and $Q_{ext}(x,\gamma)$ is the extinction efficiency factor expressed in $\text{m}^{-1}$ for the size parameter $x$ and the refraction index ratio $\gamma = n_1/n_2$.

The extinction coefficient $k(\lambda)$ therefore depends on the wavelength but also on the DSD. To study the influence of fog particle size and wavelength on the extinction phenomenon, $Q_{ext} = f(\lambda)$ and $Q_{ext} = f(d)$ with $d$ the droplet diameter are plotted.

The $Q_{ext} = f(\lambda)$ plot (Figure 3a) shows a maximum extinction peak for each diameter $d$. The position of this peak depends on the diameter. For particles with larger diameters, this extinction peak moves towards the infrared. Extinction is very important in the visible domain for water droplets with a diameter between 1 μm and 2 μm. For droplets with a diameter greater than 2 μm, extinction is maximum in the infrared. This will help us to discuss the experimental results. The plot $Q_{ext} = f(d)$ in Figure 3b also highlights the extinction peak and shows that curves tend to the value of $Q_{ext} = 2 \text{ m}^{-1}$.

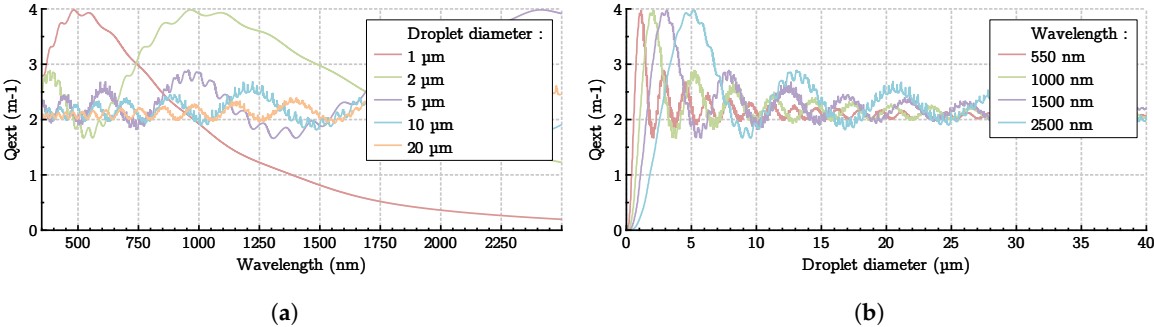

**Figure 3.** Extinction efficiency factor of spherical particles according to Mie theory. (**a**) As a function of wavelength; (**b**) as a function of the diameter.

Equations (3)–(9) were implemented in C++ to determine the extinction coefficient for each wavelength from the DSD measurements. We then refer to the Mie model in the following.

As the Mie model remains partial (without molecular impact and in the context of single scattering), MODTRAN software [9] was also used to avoid these two latter assumptions. This moderate resolution atmospheric transmission model solves the radiative transfer equation [14] by computing line-of-sight (LOS) atmospheric spectral transmittances and radiances for participating media (molecular and clouds):

$$\frac{\partial L_\lambda}{\partial u}(r,u) = -k(\lambda)L_\lambda(r,u) + \beta(\lambda)\frac{1}{4\pi}\left(\int_{4\pi} L_\lambda(r,u_i)\phi_\lambda(u_i,u)d\Omega_i\right), \qquad (10)$$

where $L_\lambda(r,u)$ is the spectral radiance for the wavelength $\lambda$ at point $r$ and in direction $u$, expressed in $\text{W·m}^{-2}\text{·sr}^{-1}\text{·m}^{-1}$, and $d\Omega_i$ is the elementary solid angle. The terms $\beta(\lambda)$ and $k(\lambda)$ are respectively the scattering coefficient and the extinction coefficient given by the Mie theory. The function $\phi$ is the scattering phase function making it possible to take into account energy gain by in-scattering [14]. In the MODTRAN model, the medium is modeled in one dimension: it is assumed to be homogeneous in the other two dimensions. The software computes the spectral radiance received by an observer from the black body radiance emitted by a source. Users set the source temperature, the Mie coefficients of the participating media (including the scattering phase function), and the distance between the source

and the observer. In this work, we used the black body source, whose spectrum is plotted in Figure 4, and a molecular composition of a typical continental atmosphere at ground level. It is important to note that when we omit the second term of the right-hand side of the Equation (10), it goes back to our Mie model.

This first section presented the most common modeling of the propagation of light in fog. It takes into account DSD and wavelength to estimate an extinction coefficient. It is also important to look at direct measurements of this extinction coefficient.

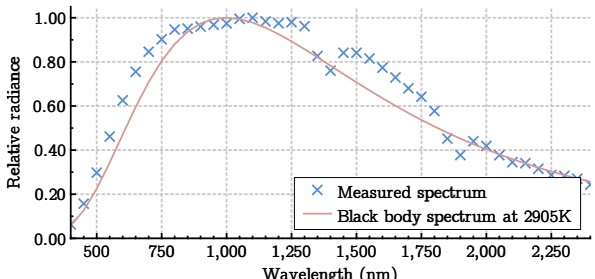

**Figure 4.** Spectrum of the 400-W halogen lamp used as a light source.

## 3. Materials and Methods

### 3.1. Pavin BP: A Fog and Rain Facility

This study was made possible by using the PAVIN BP "Fog and Rain" platform for intelligent vehicles. This platform was developed to investigate all transport systems that could be affected by adverse weather such as fog and rain. The PAVIN BP platform, shown in Figure 5, is an infrastructure allowing fog and rain to be generated in controlled conditions. This 31 m-long platform includes a 15-m fixed section (tunnel) and a 16-m greenhouse with a transparent envelope for daytime conditions and an opaque cover for night time conditions. This platform is 5.5 m wide and 2.3 m high. An observation post is located at the end of the platform, in order to install the equipment outside of the wet area during testing. Fog is produced by nozzles spraying water under high pressure. It is therefore possible to produce fog of different densities by modifying the quantity of water injected, in all temperature and humidity conditions. Visibility (MOR) is measured by a transmissometer and can be kept constant by 10-m increments between 10 m and 100 m. Different DSDs can be produced by changing the nature of the water injected (normal tap water or demineralized water). These are similar to those obtained for some natural fogs as measured by a PSA [17].

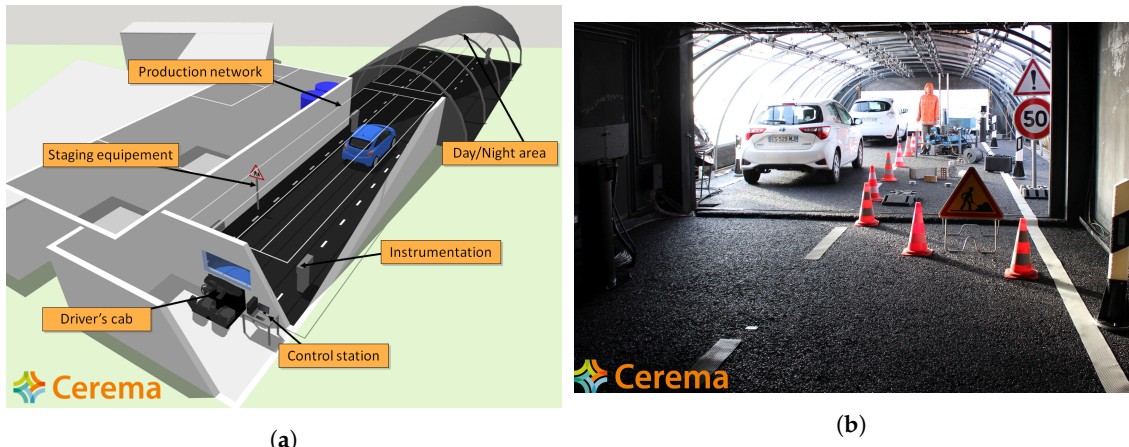

(**a**)  (**b**)

**Figure 5.** The PAVIN BP platform. (**a**) Exploded diagram of the infrastructure, presenting the 3 parts: the observation post, the tunnel section for night conditions, and the greenhouse section for day conditions; (**b**) day view with cars, work zone, and various signaling equipment.

In this infrastructure, methodologies to compare artificial and human vision systems can be developed, and the performance of dedicated sensors for driving assistance systems and vision sensors (cameras, LiDARs, and radars) used for the development of perception systems for future autonomous vehicles can be measured. Various driving situations can be reproduced with realistic targets and also reference reflectance-calibrated targets.

### 3.2. Methodology

The measurement protocol was set up in Cerema's PAVIN BP platform, presented in the previous section. It used the following equipment:

- a Spectral Evolution PSR+3500 spectroradiometer measuring radiance for each wavelength from 360–2450 nm. Radiance measurements were grouped into 10-nm packets;
- a continuous spectrum light source. The light source chosen was a 400-W halogen lamp whose characteristics were measured. This source was stable over time, and its relative spectrum was measured and is shown in Figure 4. The halogen source was similar to a black body whose temperature was estimated at 2905 K by optimization; this value was used in the MODTRAN model;
- a PALAS WELAS 2100 PSA measuring the DSD of fog over the range of 0.3–17 μm. This PSA identified the number of drops in 60 classes in this range;
- a Degreane Horizon TR30 transmissometer measuring the MOR throughout the measurements.

Throughout the tests, the sensors were all placed in the same horizontal plane, 1.20 m above the ground. The timestamps of all sensors were synchronized.

The light source was placed in front of the spectroradiometer at a distance $D$ (see Figure 6). The spectroradiometer was positioned to target the light source. To prevent any condensation on the optics of the spectroradiometer, the latter was heated. The values of $D$ varied from 2.5 m to 27.5 m in 2.5-m steps. The PSA was placed near the light source/spectroradiometer axis in order to limit errors due to the imperfect homogeneity of the fog. It was then calibrated.

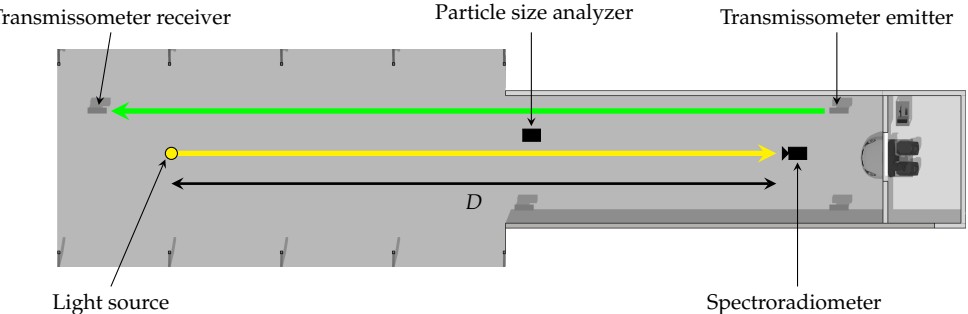

**Figure 6.** Protocol used to measure the extinction coefficient on the PAVIN BP platform.

All measurements were made in the dark to avoid intrusive light.

For each distance $D$:

- a reference measurement was made with the spectroradiometer with no fog (for an MOR greater than 1000 m, according to the WMO standard [10]);
- the fog was produced in the facility until it reached an MOR of 10 m (very dense fog). The minimum value of 10 m was chosen because this was the limit of the transmissometer measurement;
- the production of fog was stopped, and measurements were then made continuously as the fog dissipated from 10 m–1000 m. The measurements included that of the MOR using the transmissometer, the radiance received by the spectroradiometer, and the DSD measured by the PSA;

- to obtain enough data, a second complete dissipation was reproduced using the same protocol.

After making measurements for all distances $D$ with the first reproducible fog type on the platform (radiation fog, with a 0.5-µm centered radius mode), measurements were made with the second fog type (advection fog, with two radius modes centered at 0.5 and 5 µm) for each distance $D$.

During measurement, the actual DSD of the fog was recorded with one-second steps. The DSD was then used as input for the two models. Figure 7 presents the DSD recorded during the measurement for each MOR. The MOR was here given by the transmissometer; the MOR calculated from DSD data might be different. In particular, the DSD was truncated because the device used during the experiment did not allow particles above 17 µm to be detected, as shown in Figure 7. However, the largest drops had a strong impact on visibility estimation. In Figure 7, the DSD is presented in log scale, which is more common in the meteorological domain. This explains why the standard deviation represented was visually asymmetrical. Figure 7 shows that the standard deviation was high. There were several reasons for this. Firstly, measurements were made with one-second steps, without temporal smoothing. This choice was made so as to be able to measure DSD during fog dissipation, which may take place quickly, especially for high MOR. Secondly, DSD measurement was difficult because it was based on the continuous suction of a small sample of fog. This measurement generally made sense on data smoothed out over several tens of seconds. Nevertheless, in our case, the mean DSD value made sense, because the latter did not change, whatever the temporal smoothing of the data. Finally, the small sampling volume of the PSA (a few cm$^3$), compared to the reference sensor for visibility measurements (several liters) may also partly explain the large standard deviation. To limit this, particular care was taken to position the PSA as close as possible to the transmissometer and spectroradiometer measurement areas.

During the tests, carried out from 6 December 2018 to 14 January 2019, 6754 spectra were measured with the spectroradiometer and 25,590 DSDs were measured with the PSA. For greater clarity, we limited the MOR presented in this paper to the following list: 15, 20, 30, 40, 50, 70, 100, 125, 150, and 175 m. Only 4456 of 6754 spectra were then used. As the data from the various sensors were perfectly synchronized, the data from each of them could then be processed simultaneously.

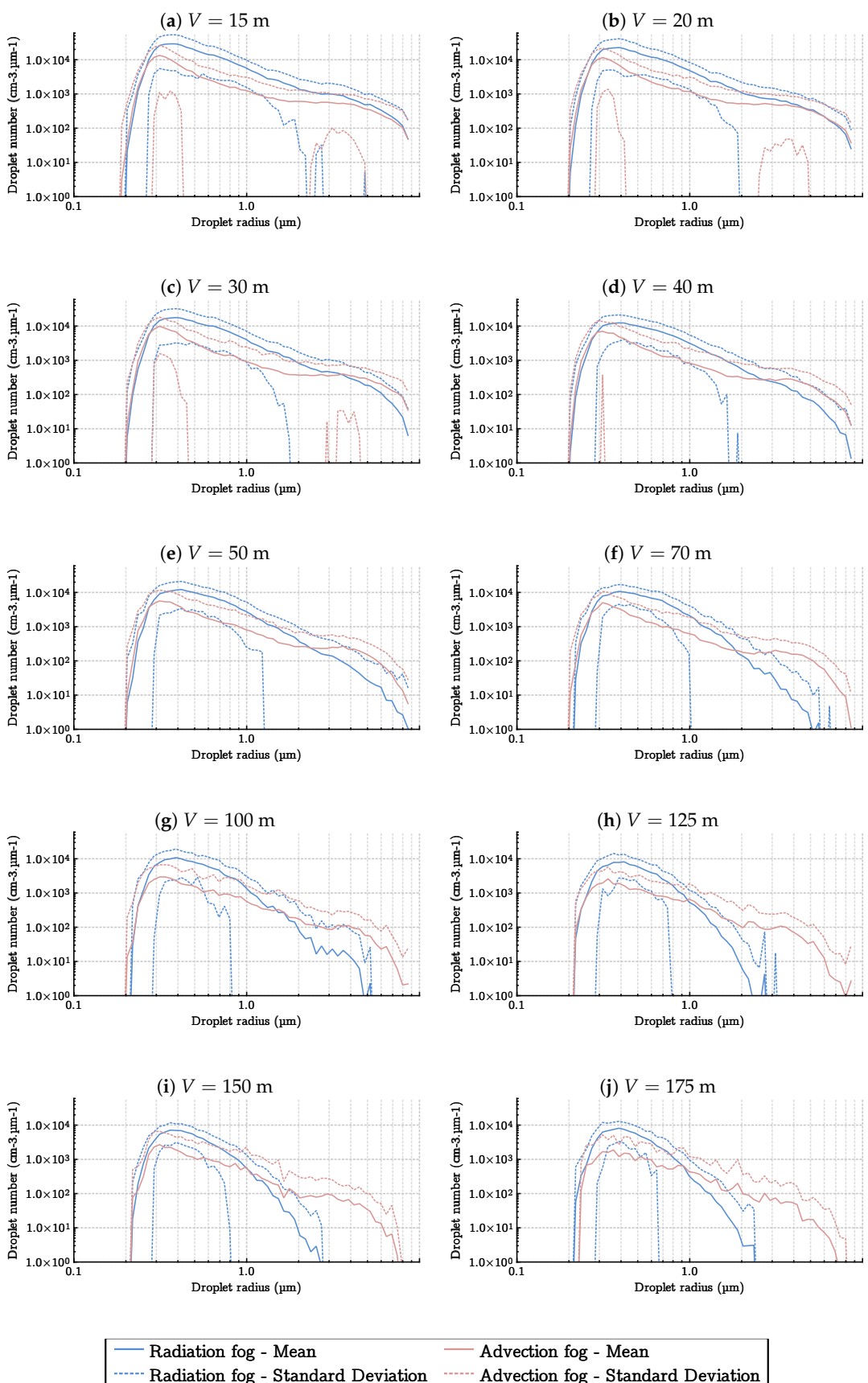

**Figure 7.** Droplet size distributions measured in the PAVIN BP facility for each MOR.

## 4. Extinction Coefficient Results

### 4.1. Estimation of the Extinction Coefficient

Three devices measuring three different quantities were used during the tests carried out. In order to perform the analysis, it was necessary to link the data of the three sensors by their time stamp. A database containing the following information was created:

- spectroradiometer/light source distance $D$,
- fog type (advection or radiation),
- MOR (temporal resolution of one second),
- DSD (temporal resolution of one second),
- and radiance for each wavelength $\lambda$ (temporal resolution of a few seconds).

For both types of fog, the radiance as a function of wavelength $\lambda$, MOR, and distance $D$ could therefore be obtained. The radiance is therefore expressed in the following form:

$$\text{Radiance} = Rad(\lambda, V, D) \tag{11}$$

The measurements were then grouped by MOR, in increments of 5 m (plus or minus 2 m) from 10 m–200 m (10, 15, 20, ..., 200 m).

For a fixed spectroradiometer/light source distance and a fixed fog type, several measurement points were obtained. However, this number of measurements was not constant as a function of MOR. This was because fog dissipated slowly in the platform when MOR was low (dense fog), and then, the rate of fog dissipation increased with increasing MOR (light fog). This is shown in Table 1. A sufficient number of measurements was still obtained for all the cases tested.

**Table 1.** Statistics on the number of measurements made during the tests. Spectroradiometer data were collected at a variable time step depending on the conditions (a few seconds). The PSA data were collected at a time step of one second.

| MOR, $V$ (m) | Number of Spectroradiometer Measurements | | Number of PSA Measurements | |
|---|---|---|---|---|
| | Radiation Fog | Advection Fog | Radiation Fog | Advection Fog |
| 15 | 360 | 238 | 2198 | 1427 |
| 20 | 240 | 206 | 1399 | 1185 |
| 30 | 241 | 121 | 1354 | 688 |
| 40 | 214 | 96 | 1057 | 567 |
| 50 | 141 | 100 | 732 | 519 |
| 70 | 76 | 90 | 320 | 480 |
| 100 | 57 | 34 | 202 | 176 |
| 125 | 64 | 24 | 266 | 133 |
| 150 | 58 | 30 | 235 | 127 |
| 175 | 54 | 27 | 250 | 112 |
| Ref (>1000 m) | 1028 | 957 | 0 | 0 |
| Total | 2533 | 1923 | 8013 | 5414 |

In order to determine the extinction coefficient $k(\lambda)$ for each wavelength, the following procedure was applied.

Owing to the fact that the spectroradiometer resolution was 8 nm for wavelengths between 1000 nm and 1900 nm, all spectral measurements were grouped every 10 nm. Furthermore, in order to reduce the database, MORs were grouped every 5 m. Each MOR class (e.g., the class for 10 m) contained the MOR value of the class, as well as those within plus or minus 2 m of this value (e.g., [8; 12 m]).

All radiance spectra measured by the spectroradiometer for a three-tuple "source/receiver distance", "fog type", and "MOR class" were collected and sorted.

After recovering the raw spectrum, the ratios of the radiance in fog condition versus the reference radiance at the same distance $D$ were calculated using Equation (12). As in the previous step, a spectrum ratio was obtained for a fixed distance $D$, fog type and MOR. These ratios were then used to determine the extinction coefficient corresponding to a given MOR and wavelength.

$$Ratio(\lambda, D, V) = \frac{Rad(\lambda, D, V)}{Rad_{ref}} \tag{12}$$

where $Rad_{ref}$ is the mean of $Rad(\lambda, D, V)$ with $V \geq 1000$ m.

From the ratios previously calculated, it is now possible to go back to the value of the $k$ coefficient for each pair $(\lambda, V)$. This is possible using the relationship:

$$Ratio(\lambda, D, V) = e^{-k(\lambda, V).D} \tag{13}$$

Therefore, to calculate the extinction coefficient $k$, it is sufficient to calculate the slope $a$ of the following function by orthogonal linear regression with a vanishing intercept coefficient:

$$D \to ln(Ratio(\lambda, D, V)) = a.D \tag{14}$$

Then,

$$k(\lambda, V) = -a. \tag{15}$$

The extinction coefficient depends on the MOR $V$ and the wavelength $\lambda$. This operation must therefore be performed for each type of fog, each wavelength, and each MOR. An example of regression is given in Figure 8.

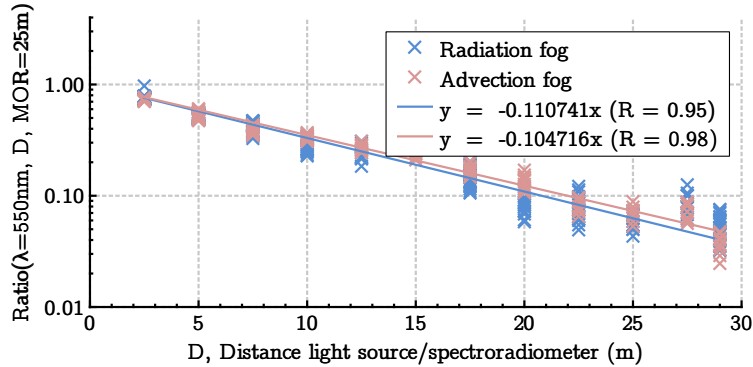

**Figure 8.** $Ratio(\lambda, D, V)$ as a function of $D$ for $\lambda = 550$ nm and $V = 25$ m for the two types of fog generated in the PAVIN BP facility.

In Figure 8, the two curves give the same value of the extinction coefficient $k = 0.11$ m$^{-1}$ (resp. $k = 0.10$ m$^{-1}$) for radiation fog (resp. advection fog). It is also possible to calculate the reference extinction coefficient $k_{ref}$ from the MOR in meters, measured by the transmissometer, which was considered the reference sensor. This was true only for wavelength $\lambda = 550$ nm, used by the transmissometer for the measurement, with the following relationship:

$$k_{ref} = -\frac{ln(0.05)}{V} \tag{16}$$

The numerical application with $V = 25$ m gave $k_{ref} = 0.12$ m$^{-1}$. In the case of Figure 8, the reference value of $k_{ref}$ therefore corresponded approximately to the values measured by the spectroradiometer. However, the data for the two types of fog did not perfectly follow the theoretical

model. The correlation coefficient $R$, between zero and one, made it possible to make a synthetic measurement of the reliability of the relationship between $D$ and $ln(Ratio(\lambda, D, V))$. The closer $R$ was to one, the better the estimation of $k$.

Table 2 gives the correlation coefficient as a function of MOR. As shown in Table 2, the correlation coefficient globally decreased as the MOR increased. This was due to the fact that there was more instability when the MOR increased because fog dissipation was faster for a higher MOR (see Table 1). The correlation coefficient was also unavailable (NA values) for $\lambda = 360$ nm and 2400 nm and $V = 15$ m: this was due to the spectroradiometer measurement range, which did not have enough light signals. These wavelengths were at the limit of the spectroradiometer, and the sensitivity was therefore lower for these. When the correlation coefficient was lower than 0.9 ($R \leq 0.9$), the measurements were labeled as uncertain data. The value of 0.9 was arbitrarily chosen, but makes Figure 9 easier to read.

**Table 2.** Correlation coefficient $R$ of the linear regression (14) for five wavelengths.

| MOR (m) | | Radiation Fog | | | | | Advection Fog | | | | |
|---|---|---|---|---|---|---|---|---|---|---|---|
| | $\lambda =$ | 360 | 550 | 1000 | 2000 | 2400 | 360 | 550 | 1000 | 2000 | 2400 |
| 15 | | NA | 0.97 | 0.97 | 0.77 | NA | 0.96 | 0.98 | 0.98 | 0.64 | NA |
| 20 | | 0.91 | 0.98 | 0.98 | 0.99 | 0.96 | 0.96 | 0.98 | 0.98 | 0.98 | 0.66 |
| 30 | | 0.94 | 0.96 | 0.97 | 0.98 | 0.97 | 0.97 | 0.98 | 0.99 | 0.99 | 0.98 |
| 40 | | 0.93 | 0.94 | 0.96 | 0.97 | 0.96 | 0.97 | 0.99 | 0.99 | 0.99 | 0.98 |
| 50 | | 0.90 | 0.90 | 0.93 | 0.93 | 0.93 | 0.94 | 0.97 | 0.97 | 0.98 | 0.98 |
| 70 | | 0.84 | 0.84 | 0.87 | 0.88 | 0.87 | 0.93 | 0.96 | 0.97 | 0.97 | 0.97 |
| 100 | | 0.76 | 0.79 | 0.81 | 0.79 | 0.76 | 0.80 | 0.92 | 0.94 | 0.96 | 0.95 |
| 125 | | 0.49 | 0.50 | 0.67 | 0.78 | 0.78 | 0.69 | 0.85 | 0.88 | 0.92 | 0.91 |
| 150 | | 0.73 | 0.70 | 0.72 | 0.67 | 0.64 | 0.64 | 0.83 | 0.87 | 0.91 | 0.91 |
| 175 | | 0.71 | 0.70 | 0.74 | 0.67 | 0.66 | 0.61 | 0.80 | 0.89 | 0.91 | 0.88 |

The extinction coefficient $k$ was calculated for each wavelength: it is now possible to examine the influence of wavelength on this coefficient.

### 4.2. Analysis of the Extinction Coefficient as a Function of Wavelength

The influence of wavelength on the extinction coefficient was examined by analyzing Figure 9, representing $k(\lambda, V)$ as a function of $\lambda$ for different MORs from 15 m up to 175 m. Each sub-figure of Figure 9 contains two graphs constructed from spectroradiometer measurements and the method proposed in the previous section: one corresponding to radiation fog (blue scatter-style crosses and circles) and the other corresponding to advection fog (red scatter-style crosses and circles) produced in the PAVIN BP facility. Two graphs were added to each sub-figure of Figure 9 from the data measured by the PSA for both types of fog (simple dashed line for the Mie model). The PSA data were pre-processed using the Mie formulas in Section 2.3. The Mie theory proposed in the previous section did not take into account molecular absorption and scattering or the phase function in the radiative transfer function. The data from the PSA were then also processed with MODTRAN to estimate the extinction coefficient with greater accuracy [9]. The corresponding two graphs were then added to each sub-figure of Figure 9 (continuous line for the MODTRAN model). For each of these, the reference extinction coefficient $k_{ref}$ at $\lambda = 550$ nm was calculated from the MOR measured by the transmissometer, considered as a reference. This coefficient $k_{ref}$ was used to normalize all the curves, in order to have the same extinction coefficient for 550 nm. This choice was made by MODTRAN and had the advantage of focusing more closely on the influence of wavelength, particularly for analyzing data from the PSA, without taking into account the total concentration of fog droplets. Many conclusions can be drawn from these graphs.

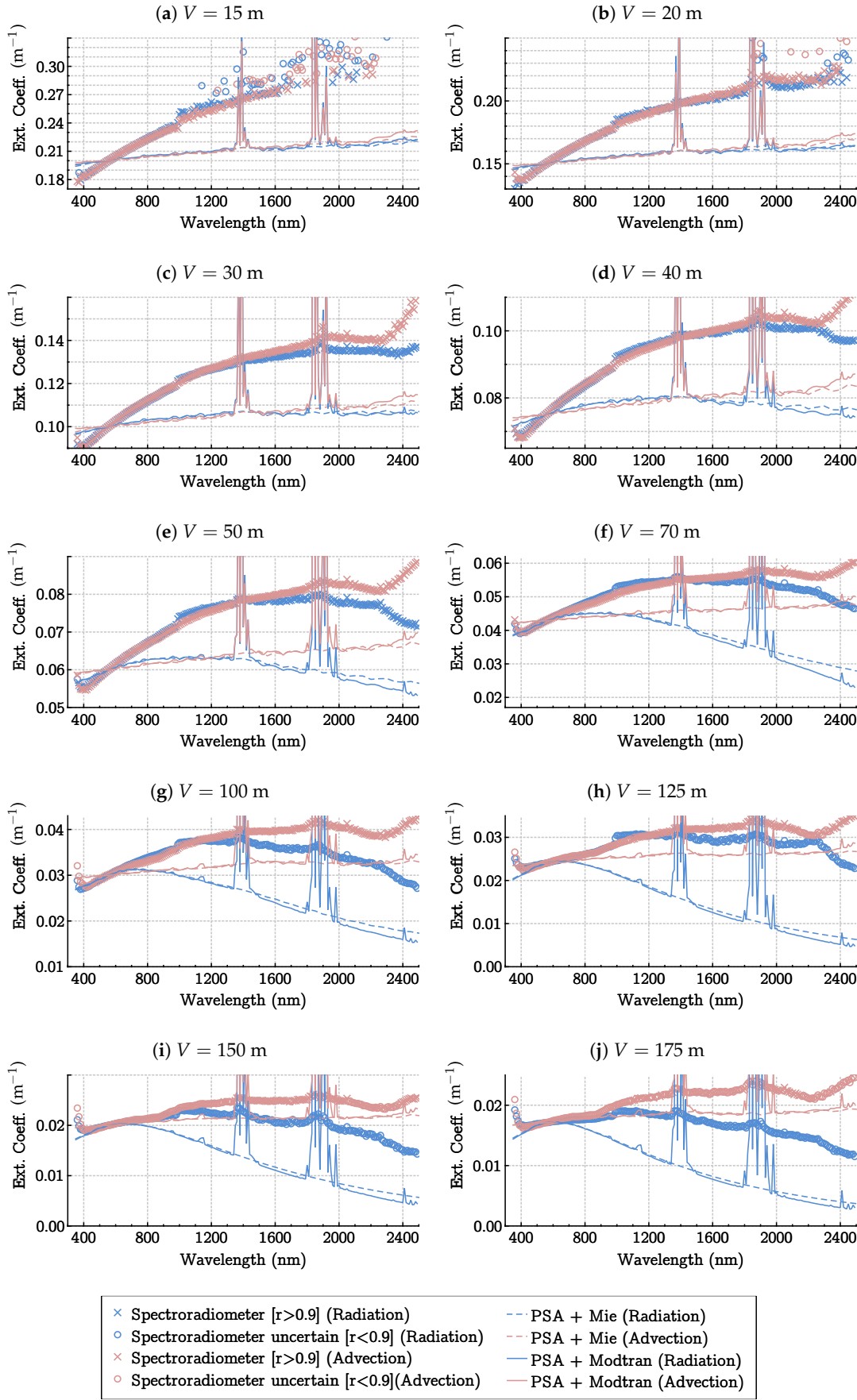

**Figure 9.** Extinction coefficient ($k$) as a function of wavelength ($\lambda$) for different MORs ($V$). In all the sub-figures, the grid has constant steps of 0.01 m$^{-1}$.

First of all, it can be noted that for both types of fog (advection and radiation), the spectroradiometer measurements (scatter-style crosses and circles) have the same behavior as the curves from the two models + PSA measurements (simple dashed line for the Mie model and continuous line for the MODTRAN model). This shows that the two methods and the sensors used agreed, showing differences in the extinction coefficient according to wavelength, MOR, and fog type. However, the model + PSA combination would tend to underestimate systematically the extinction coefficient compared to spectroradiometer measurement. This occurred especially for the densest fogs ($V < 100$ m). This may be due to the fact that the PSA would miss drops when confronted with extremely dense fogs. This was because the sensor performed filtering to prevent several drops from being counted at the same time. In the case of very dense fogs, these filters may have an impact on the measurement. Moreover, at wavelengths of 440–1600 nm and $V < 125$ m, the extinction coefficient from the spectroradiometer measurements varied more than the extinction coefficient from the two models + PSA measurements. The PSA may underestimate fog droplets with a diameter between 2 and 5 μm and overestimate droplets with a diameter higher than 5 μm (see Figure 3a). In the case of radiation fog (blue curves) and an MOR greater than 100 m, underestimation of the extinction coefficient by the model + PSA measurement may also result from the fact that the sensor cannot detect drops whose diameter is less than 0.3 μm. Again, all the drops may not be counted.

Concerning the relationship between the extinction coefficient and the wavelength, it appears that the extinction coefficient clearly depends on the wavelength. In particular, the extinction coefficient was higher for near-infrared wavelengths than for visible wavelengths, particularly for the densest fog ($V < 30$ m). The ratio can reach 1.5 for an MOR of 15 m. This means that dense fogs have a greater impact on the near-infrared than on the visible domain. For lighter fogs, the difference becomes smaller (ratio less than 1.25). As a reminder, by definition, the extinction coefficient is considered constant in the visible range (350 nm–800 nm) as the MOR is based on human perception [10], but the latter had a ratio of 1.30 on the curves. In the particular case of light advection fog (MOR higher than 100 m), a wavelengths above 2200 nm had a lower attenuation coefficient than the visible one: this was due to the fact that in this case, the drops were very small (lower than 1 μm). The longer wavelengths were less affected, as previously shown in Figure 3a.

Concerning the comparison between the two models + PSA measurements (Mie and MODTRAN), it can be said that the two models were quite similar. However, the MODTRAN model tended to underestimate globally slightly the extinction coefficient compared to the Mie model, except for the molecular scattering peaks. This was due to the assumption of multiple scattering made by MODTRAN (as opposed to the assumption of single scattering made by the Mie model) [5]. On the other hand, it was interesting to see that the complex MODTRAN model made it possible to highlight extinction peaks around the wavelengths of 1340 nm–1440 nm and 1910 nm–1950 nm. This was due to the molecular absorption of $H_2O$, as we can see in Figure 10. Spectroradiometer measurements also showed the same instability in these areas. The latter had a much smaller amplitude than for MODTRAN + PSA measurements. However, this can be explained by the fact that the values were integrated over ranges of 10 nm in the case of the spectroradiometer. This result confirmed previous work [6].

Concerning the impact of the type of fog, it appears that advection and radiation fogs had exactly the same impact on light at any wavelength from 350 nm–1000 nm, for all MORs. This was in accordance with the literature, since whatever the type of fog, the definition of the extinction coefficient is the same. For the densest fogs ($V < 40$ m), the extinction was the same for both types (advection and radiation). This result was confirmed by the fact that the DSDs were similar for both types of fog below an MOR of 40 m, as shown in Figure 7. On the other hand, above 1000 nm, and for less dense fog ($V > 40$ m), the extinction coefficient was lower for radiation fog than for advection fog. Radiation fog produced in the PAVIN BP facility therefore had less impact on light than advection fog for wavelengths above 1000 nm. This can be explained by the presence of small fog droplets only (with a diameter of less than 1 μm), which had little impact on the highest wavelengths. The results obtained

with the spectroradiometer were then confirmed by the model + PSA measurements. The impact measured was around 10%.

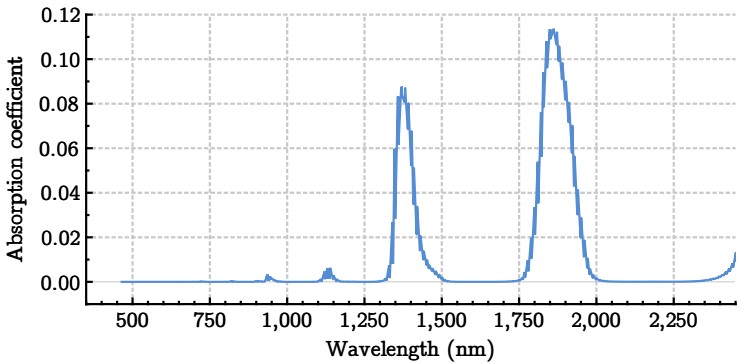

**Figure 10.** $H_2O$ absorption spectrum, with a wavelength step greater than 0.5 nm.

## 5. Discussion

This study showed that the literature models (here Mie and MODTRAN) and direct measurements of the extinction coefficient showed the same trends even if there was a difference between them. On the other hand, a limit to the use of the Mie model was shown, which is, however, the most widespread. More complex models such as those used in MODTRAN may reveal local behavior. For example, MODTRAN solves the radiative transfer equation including the radiative energy gain due to in-scattering (involving the phase function of the Mie theory) and molecular scattering, which can lead to local spectral behavior. These complementary elements to the Mie model (which takes into account only single particle scattering according to Mie theory) were important in refining the model. The question then arises as to which elements have the greatest impact in refining the model. This may be the subject of further study.

Wavelengths from 350 nm to 1000 nm had the same behavior in fog, regardless of fog density or type. On the other hand, above 1000 nm, differences may occur. For very dense fogs ($V < 30$ m) and in particular advection fogs, fog had an impact about 10% higher for wavelengths in the near-infrared range (1000 nm–2400 nm) than for visible wavelengths (400 nm–800 nm). This kind of fog has larger drops, having a greater impact at higher wavelengths. On the other hand, for light radiative fog ($V > 100$ m), the impact of the fog was equal, or even lower, for wavelengths above 1000 nm. For light fog, therefore, we have the same conclusion as that given in the literature [2]. This is very positive and confirms the validity of the work presented. It is therefore essential to take into account the DSD of the fog when comparing extinction coefficients above 1000 nm. This calls into question the use of some models of the extinction coefficient as a function of wavelength proposed in the literature [3] without verifying the similarity of the DSD. However, some models [4] give several possible DSDs, which is a good point.

Regarding possible tests of near-infrared sensors in fog conditions within the PAVIN BP platform, it is essential to always test these systems on advection fog or very dense radiation fog ($V < 30$ m). These fogs contain larger droplets, at least larger than 2 μm. These droplets have a much higher impact on the near-infrared domain than droplets with a diameter of 1 μm or less.

Two main questions are still pending and deserve to be addressed in further research. Firstly, it seems important to compare the advection and radiation fogs produced on the PAVIN BP platform with natural fogs. This study showed the importance of the DSD on the extinction coefficient for near-infrared wavelengths. It is then necessary to measure the DSD of natural fogs and compare it to that of fogs produced in the PAVIN BP platform. Secondly, the good correlation between the measurements of the coefficient correlation made with a spectroradiometer and the combination model + PSA suggests that we might consider posing the problem the other way around. It would then be

possible to use the spectroradiometer as a PSA, by optimizing a theoretical DSD to obtain the measured extinction coefficient as a function of wavelength.

Concerning LiDAR technologies, it appears that the choice of wavelength, 905 nm vs. 1550 nm, should not be justified by the fog criterion. This is because these two wavelengths have close extinction coefficients (systematically, a ratio between 1.0 and 1.1), while the eye-safe emitted power is at least 20-times greater in the case of 1550 nm [1]. It should be noted, however, that care should be taken to avoid the ranges 1340 nm–1440 nm and 1910 nm–1950 nm, which have been identified as potentially disturbed due to strong $H_2O$ molecular absorption (according to MODTRAN modeling). These conclusions are the same for time-of-flight cameras and for other active sensors working in the near-infrared spectral band.

To conclude, direct spectroradiometer extinction measurements and extinction estimation using models (Mie and MODTRAN) taking into account the actual DSD showed the same trend even though they were different. The results obtained suggest that it would be possible to use a spectroradiometer measurement to estimate the DSD. Finally, the choice of using the wavelength 905 nm vs. 1550 nm in LiDARs should not be made on the basis of fog conditions: there is a small difference (<10%) between the extinction coefficients at these two wavelengths for the same emitted power in fog.

**Author Contributions:** Conceptualization, P.D., M.C., and F.B.; data curation, P.D.; formal analysis, P.D.; funding acquisition, M.C.; investigation, P.D.; methodology, P.D.; project administration, M.C.; software, P.D. and F.B.; supervision, F.B.; validation, P.D., M.C., and F.B.; visualization, P.D.; writing, original draft, P.D., M.C., and F.B.; writing, review and editing, P.D. and F.B.

**Funding:** The research leading to these results has received funding from the European Union under the H2020 ECSELProgramme as part of the DENSEproject, Contract Number 692449. This work has been sponsored by the French government research program "Investissements d'Avenir" through the IMobS3Laboratory of Excellence (ANR-10-LABX-16-01) and the RobotExEquipment of Excellence (ANR-10-EQPX-44), by the European Union through the Regional Competitiveness and Employment program, 2014–2020 (ERDF–AURAregion), and by the AURA région.

**Acknowledgments:** The authors would like to thank Jean-Luc Bicard and Antony Bayle for their active participation in this publication.

**Conflicts of Interest:** The authors declare no conflict of interest. The funders had no role in the design of the study; in the collection, analyses, or interpretation of data; in the writing of the manuscript; nor in the decision to publish the results.

## Abbreviations

The following abbreviations are used in this manuscript:

| | |
|---|---|
| MOR | Meteorological optical range |
| DSD | Droplet size distribution |
| WMO | World Meteorological Organisation |
| FSO | Free space optics |
| TRL | Technology readiness levels |
| SWIR | Short wave infrared |
| ADAS | Advance driving assistance systems |
| MODTRAN | Moderate resolution atmospheric transmission |
| LOS | Line-of-sight |

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
