# Peer review of "Light Transmission in Fog: The Influence of Wavelength on the Extinction Coefficient"

_applsci, doi:10.3390/app9142843_

Round 1
Reviewer 1 Report
The comments in details are given as following:
1. Page-1, Line 13-14. The contents are not yet demonstrated in this paper, please remove them from the Abstract. It may be put in the Discussion section.
2. Page-5, Eq.(9). When you use the constant of 1e+5 in the formula, please give the unit for each variable.
3. Page-6, Eq.(10). Please describe the variable “k(λ)” and “Ω”. There is no molecular absorption?
4. Page-8, Figure 6. It is surprising that there are no measurements of air temperature, relative humidity and pressure, which are important for the fog generation.
How to control the two types of fogs generation?
For the granulometer, spectroradiometer and transmissionmeter, please give the temporal resolution of the instrument and measurement at each MOV or range ‘D’.
5. Page-12, Table-2. At the MOR=125-m for the radiation fog, why are the correlation coefficient R much lower?
How long can the fog or its DSD (droplet size distribution) be stablized? You can calculate the effective radius of the fog and plot their temporal variability during the measurements.
6. Page 14, Figure 11. This figure shows the major results in this study. Two main questions:
At the wavelength of 440-1600nm and V<125 m, the extinction coefficients from the spectroradiometer measurements and the Mie or Modtran calculation show the different spectral curvatures or wavelength dependence. Why?
For the V>50-m, the extinction coefficients calculated from the Mie and Modtran show huge differences at the wavelengths with the H2O absorption. Why?
7. Page-15, Line 367-368. I don’t see the good agreement of fog extinction coefficients between the Mie (Modtran) model and the direct measurements. In fact, the Fig.11 shows different magnitude and spectral dependence.
8. Page-16, Line 404-406. These conclusions are not convinced. The Fig.11 doesn’t show their good agreements between the model simulations and the direct measurements.
Author Response
General response
First, the authors would like to thank reviewers for the time they devoted to reviewing, but also for the relevance of the comments they made.
The two major comments of the three reviewers were:
- the poor introduction and the weird part “State of the art”: to answer to this totally justified comment, we merge both part into a new introduction, more precise.
- the absence of temporal data or statistic about DSDs, and a possible misunderstanding that the DSD has not been measured for each MOR: we added all the DSD measurements, with standard deviation in the paper. Then this point is clearer.
To consider all my following comments, be careful: figure 4 and 5 where merged, and figure 2 and 3 were merged. The former figure 9 is now figure 7 and the former figure 11 is now figure 9.
Specific responses
1. Ok, the modification was done.
2. Ok, missing units were added.
3. Page-6, Eq.(10). “k(λ)” and “Ω” descriptions where added. Yes, as mentioned in the paper there is molecular absorption, in MODTRAN model.
4. « Fog is produced by nozzles spraying water under high pressure. Thus, it is possible to produce fog of different densities by modifying the quantity of water injected, in all temperature and humidity conditions. Visibility (MOR) is measured by a transmissometer and can be kept constant by increment between 10 m and 200 m. Different DSD can be produced by changing the nature of the water injected. These ones are similar to those obtained for some natural fogs as measured by an optical granulometer \cite{Colomb2008}.» added in Page 7 line 201
As the process to produce fog is mechanical and not thermodynamical, it can be produced at any pressure and temperature (concerning the relative humidity, it is always above 95%). We have sensors to do the measurements of temperature and humidity, but only if needed by the experiment (not for fog control).
See Colomb et al. 2008 for more information about the two fog conditions.
« MOR (temporal resolution of one second),
DSD (temporal resolution of one second),
and radiance for each wavelength \lambda (temporal resolution of a few second). » added in Page 9 Line 275
5. Page-12, Table-2. At the MOR=125-m for the radiation fog, why are the correlation coefficient R much lower?
You’re right, there are much lower correlation coefficient at the MOR 125m. There are local variations related to the measurement. Since there is less data when fog is less dense (higher MOR), there may be instabilities in the measurements, which are identified by the correlation coefficient. In order to be transparent, we have left all the data and clearly display the quality of the measurements through the correlation coefficient.
How long can the fog or its DSD (droplet size distribution) be stablized? Mean DSD and standard deviation where added in new Figure 7, for each MOR.
6. Page 14, former Figure 11, now figure 9:
« Moreover, at the wavelength of 440 - 1600 nm and V < 125 m, the extinction coefficient from the spectroradiometer measurements varies more than the extinction coefficient from two model + granulometer measurements. The granulometer may underestimate fog droplets with a diameter between 2 and 5 \mu m and overestimate droplets with diameter higher than 5 \mu m (see Fig.~\ref{FIG_qExt_lambda}). » added page 13, line 355.
« On the other hand, it is interesting to see the complex MODTRAN model makes it possible to highlight extinction peaks around the wavelengths of 1340nm-1440nm and 1910nm-1950nm. This is due to the molecular absorption of H2O, as we can see in Figure 10. » on page 15, line 376.
7. Page-15, Former Line 367-368, now line 397. It was an inaccurate translation that was modified by « This study first showed that the literature models (here Mie and MODTRAN) and the direct measurements of the extinction coefficient show the same trends. »
8. Page-16, Former Line 404-406, now line 435. idem than 7.
Reviewer 2 Report
The authors of this work aim to calculate the extinction coefficient as a function of wavelength for various experimental setups with the greater goal to apply their results in autonomous driving sensors.
For this purpose, they employ both an experimental setup and two theoretical models. The methodology used is sound and the results of interest and therefore this work should be published but only after some major changes are applied.
Major comments
· The introduction is very poor and reads more like an advertisement of the DENSE project. Section 2.2 reads like an introduction but need some additions to be complete. Proper English is not used in the introduction, a problem noted throughout the manuscript.
· The authors neglect the fact that the size distribution may change over the timescales of the experiment. Are the size distributions shown in Fig 9 constant for every experiment? Do the initial size distributions remain constant over time? Please discuss these important point in a separate section. Show how much the size distribution changes over time. Additionally show how reproducible is the initial size distribution of each experiment by adding error bars (1 std) over figure 9. The observed changes may shed a light on the some discrepancies discussed but not properly addressed (see my comments below). The authors have chosen use MOR as a proxy of the size distribution. As the authors acknowledge this is an oversimplification. So adding this section is necessary to fill the gap.
· The model results do not really agree with the experimental results on Section 4.2. Some agreement is noted in visible wavelengths and at high V (MOR) values. Can the authors comment on the discrepancy? Furthermore, what is stated in lines 367-368 is contradicted by figure 11.
Minor comments
· Line 69: which modes are these. Please refer to them and explain them. “1 and 3 of [8]” is not a proper way to explain.
· Line 70: Please rephrase, instruments do not have eyes and therefore do not see anything, they include sensors that can or cannot detect (or measure) an entity (or a parameter). This is a scientific journal and slang should be avoided. Same for Line 337.
· Figure 1 and 9 should be merged and modified according to suggestions above.
· Since you are examining supermicron particles it may be worth noting that geometric optics apply as well as Mie theory, but not to the entire range.
· Sections 2.1.3 is like reading a textbook on Mie theory. Is it really needed? I would argue yes only if the authors feel that the target audience lacks expertise in Mie theory basics. Either way keep fig 4 and 5 (I suggest merging them into one figure with sections (a) and (b))
· Line 146: each abbreviation must be explained before it is used
· Avoid using the phrasing state of the art. Apart that it can be confusing, this article will be read in the future when the state of the art will be different. How do you expect for the reader to know what you mean by this phrase. Lines 167-168 do not make any sense at all. Please rephrase throughout the manuscript.
· Line 160-165: Use percentage differences instead of ratio to help the reader understand better your point
· In the lines 173-174 the authors stress the importance of size distribution in extinction coefficient measurements but only qualitatively address this issue.
· Add error bars in Fig. 9
· Line 295-296. No fewer measurements do not have an effect on correlation whatsoever. This result is not properly explained. If an experiment has 2 results , the minimum amount possible, that are statistically different, and a regression line is fitted then the correlation coefficient is by default unity. In other words the less the number of points in a fit the more likely the correlation coefficient to be high. This finding simply means that the linear rule (equation 14) is not valid. There can be a number of reasons. My estimate is that the examined system is too thick (high number concentrations) at the beginning. This would explain why fog depletes rapidly at first and slows down with time as the authors have recorded.
· Because this study aims at autonomous vehicles, I suspect only one type of fog will be regularly met. Is this true. Please discuss this point.
A kind note. Instruments with the ability to produce any size droplets are available. Atomizers have been used widely to produce submicron droplets. However, in these cases you may need to consider composition as well, a parameter completely neglected in this work and in all the given references. This comment was made based on Lines 390-395.
Author Response
General response
First, the authors would like to thank reviewers for the time they devoted to reviewing, but also for the relevance of the comments they made.
The two major comments of the three reviewers were:
- the poor introduction and the weird part “State of the art”: to answer to this totally justified comment, we merge both part into a new introduction, more precise.
- the absence of temporal data or statistic about DSDs, and a possible misunderstanding that the DSD has not been measured for each MOR: we added all the DSD measurements, with standard deviation in the paper. Then this point is clearer.
To consider all my following comments, be careful: figure 4 and 5 where merged, and figure 2 and 3 were merged. The former figure 9 is now figure 7 and the former figure 11 is now figure 9.
Specific responses
Major comments
· The introduction was modified according to your comments. Section 2.2 moved into the introduction.
· The authors neglect the fact that the size distribution may change over the timescales of the experiment. Are the size distributions shown in Fig 9 constant for every experiment? Do the initial size distributions remain constant over time? Mean DSD and standard deviation where added in Figure 7 (former fig 9), for each MOR. With these new figures, the reader has much more details on the DSD.
· More comments on the discrepancy were added. The conclusion in lines 367-368 was not well translated, it is now modified (now in line 397).
Minor comments
· Line 69 modified « … represents the models of the two most representative kind of dense fog corresponding to heavy advection fog (model 1) and heavy radiation fog (model 3) of [9]. The models 2 and 4 of [8] were not used here because they correspond to moderate fogs. ».
· Line 70 and Line 337 rephrased with « detect ».
· Since you are examining supermicron particles it may be worth noting that geometric optics apply as well as Mie theory, but not to the entire range.
· Sections 2.1.3: We think that this section is important to understand the meaning of figure 4 and 5, which are crucial for comments on figure 7 (former figure 9). Figures 4 and 5 were merged according to your comment.
· Line 146: ok.
· Avoid using the phrasing state of the art: ok, rephrased throughout the whole manuscript.
· In the lines 173-174 the authors stress the importance of size distribution in extinction coefficient measurements but only qualitatively address this issue: in our work we measure the exact DSD, then we use it as input of two models. We then adress the DSD issue qualitatively and quantitatively.
· Add error bars in Fig. 9 : Done, the standard deviation was added, now in Figure 7.
· Line 295-296. You’re right, sorry for the misspelling, we modify it: “This is due to the fact that there is more instability when the MOR increases because fog dissipation is faster for a higher MOR.”
· Because this study aims at autonomous vehicles, I suspect only one type of fog will be regularly met. Is this true. Please discuss this point: In section « 2.2 Different kinds of fog », we describe the two kind of fog. Based on your comment, we added « A land vehicle, particularly an autonomous vehicle, is likely to encounter these two types of natural fog if it moves in continental and seaside areas. »
A kind note: Thank you, we're going to take a close look at the atomizer literature.
Reviewer 3 Report
Comments concerning manuscript 484502:
These comments concern the manuscript titled: “Light transmission in fog - influence of wavelength on the extinction coefficient”. The objective seems to be determining the best sensor wavelengths for use under foggy conditions to allow obstacle detection. The “English” needs some help with missing articles, incorrect verb tense, and unclear references. The overall organization of the paper can be greatly improved as the Introduction is too brief and does not refer to descriptions of past measurements similar to what is proposed in this research (with only a minor exception). The paper then goes to a “state of the art” section which, I believe is somewhat extraneous to the manuscript. Perhaps, an outline containing the normally used headings would help in this organization. I will try to list some specific issues in the following:
Line 4-6 – these three sentences are not complete thoughts – basically thought fragments as is the remainder of the abstract.
Line 13-14 – this sentence is immaterial…
Line 21 – “the” needed before H2020…
Line23 – “the” unnecessary…
Line 28 – Lidar is singular here…
Line 29 – instead of “for both”, I think that between is better choice…
Line 30 – “it is proposed” is not what you are doing; you are making the measurements to “better understand”…
Line 34 – it is unclear whether there is more than one model used in the research – is there a model for Mie theory and another model called MODTRAN? Or, is Mie theory part of MODTRAN (it is…, I believe…) –if a separate model for the Mie theory alone is used, it should be identified…
Line 56-58 – there appears to be two paragraphs here, each 1 sentence long – there needs to be more completion of the thoughts into proper sentences and paragraphs.
Line 81 – doesn’t absorption also cause attenuation?
Line 101 – what is a granulometer? Needs to be defined…
Line 104 – there is considerable text inserted between here and line 105…some of the terms (y) are not defined…
Line 129 – sentence is extraneous or else needs more explanation…
Line 133 – what is “the problem proposed here”?
Line 146-150 – several standalone thoughts – they need to be tied together into coherent sentences
Line 158-166 – is this a review of a particular reference or ???
Line 187 – again, the word propose is used when what is meant that “it was studied”
Line 191 – using the word “thanks” here is inappropriate….
Line 280 – the word “regression” is used without further elaboration – if a regression analysis was performed, the the details should be given
Line 305 – to what does the term “the plot” refer?
Line 322 – again, there has been no description or reference for the “Mie model” and not much description of how MODTRAN was used
Figure 11 – the caption needs considerably more detail so that the plots may be understood. In addition, the word advection is misspelled in the legend…
Line 367 – this paragraph needs help – it does seem to really discuss the results and, furthermore, the statement that the research has shown limits to the use of the “Mie theory model” is not completed to say exactly what the limits are and whether the limits are due to Mie theory or the Mie theory model used.
Line 402 – what is meant by “…disturbed by MODTRAN…”?
Line 404 –“have” needs to be replaced by “has” and, perhaps, this “sentence” should be at the beginning of the “conclusion”…
Line 408 – where has the discussion of “no significance difference” been in the body of the manuscript? Furthermore, this phrase is normally attached to the results of a statistical analysis.
Author Response
General response
First, the authors would like to thank reviewers for the time they devoted to reviewing, but also for the relevance of the comments they made.
The two major comments of the three reviewers were:
- the poor introduction and the weird part “State of the art”: to answer to this totally justified comment, we merge both part into a new introduction, more precise.
- the absence of temporal data or statistic about DSDs, and a possible misunderstanding that the DSD has not been measured for each MOR: we added all the DSD measurements, with standard deviation in the paper. Then this point is clearer.
To consider all my following comments, be careful: figure 4 and 5 where merged, and figure 2 and 3 were merged. The former figure 9 is now figure 7 and the former figure 11 is now figure 9.
Specific responses
Major comment: according to your comment (and another reviewer comments), we moved the former section « State of the art » into the Introduction. Then, we have modified the introduction.
Line 4-6 – these three sentences are not complete thoughts – basically thought fragments as is the remainder of the abstract: ok done.
Line 13-14 – this sentence is immaterial: ok, removed.
Line 21, 23, 28, 29, 30: ok corrections done, thank you.
Line 34 – it is unclear whether there is more than one model used in the research – is there a model for Mie theory and another model called MODTRAN? There are two models: the Mie theory one, with Mie theory alone, and the MODTRAN one (which includes also Mie theory). The phrase was modified for better explanation.
Line 56-58 – Ok, we modified it.
Line 81 – doesn’t absorption also cause attenuation? yes as mentioned on line 136, but less important in the fog case.
Line 101 – what is a granulometer? « The DSD can be measured by a granulometer, which is an optical particle counter. » added on Line 120.
Line 104 – Missing lines number is a problem of the compiler… y is defined in the paper as the second order Bessel function.
Line 129 – ok
Line 133 – ok modified in the new introduction.
Line 146-150 – ok
Line 158-166 – it is a review of the reference 16
Line 187 – ok
Line 191 – ok
Line 280 –a regression analysis was performed for each type of fog, each wavelength and each MOR, the details are given on page 11.
Line 305, 322, Figure 9 (Former Figure 11) – ok, we added precise references to the figure and explanation at the beginning of the section 4.2.
Figure 9 (former figure 11) – correction of the legend ok
Line 367 – this paragraph needs help – ok, we added supplements.
Line 402 – sorry for the misspelling, we rephrased: « that care should be taken to avoid the ranges 1340nm-1440nm and 1910nm-1950nm which have been identified as potentially disturbed due to strong H2O molecular absorption (according to MODTRAN modelling). »
Line 404 - ok
Line 408 – where has the discussion of “no significance difference” been in the body of the manuscript?
« As a reminder, the extinction coefficient is considered constant in the visible range (350nm-800nm), but the latter has a ratio of 1.30 on the curves. »
and
« Concerning Lidar technologies, it appears that the choice of wavelength 905nm vs 1550nm should not be justified by the fog criterion. This is because these two wavelengths have close extinction coefficients (systematically a ratio between 1.0 and 1.1). »
Round 2
Reviewer 1 Report
The revised manuscript have become more concise. At this time, I have a minor comment on the discussion/conclusion about the updated Fig.7.
Figure 7 indicates that the two labeled fogs are similar with V<40-m but different with V>100-m.
This may help explain why there are large differences of the spectral variation trend and magnitude between the measurements by the spectrometer and model simulation (granulometer+Mie, granulometer+modtran). Even for ther two model simulations for the two fogs (or two measurements for the two fogs), there are still some differences.
Author Response
The authors would like to thank reviewers again for the second proofreading. The new comments that have been made are always in the right direction, and have greatly improved the article. Majors modifications are :
the term “granulometer” has been changed for “particle size analyser (PSA)” in the whole paper.
“English” improvement.
Precisions about the Mie Model.
Specific responses:
You’re right. Based on your comments, we modify the explanation between lines 383 and 393.
Reviewer 2 Report
On Line258 there is a typo make sens, should be make sense
Line 204-205 do not make sense. How can you change the nature of water? Please rephrase.
This work reads much better and should be published
Author Response
The authors would like to thank reviewers again for the second proofreading. The new comments that have been made are always in the right direction, and have greatly improved the article. Majors modifications are :
the term “granulometer” has been changed for “particle size analyser (PSA)” in the whole paper.
“English” improvement.
Precisions about the Mie Model.
Specific responses:
On Line258 : ok, thank you.
Line 204-205 : ok, we add precisions :
“Different DSD can be produced by changing the nature of the water injected (normal tape water or demineralised water).”
Reviewer 3 Report
This manuscript (#484502) is an attempt to determine the extinction coefficient of light in fog as a function of wavelength. There are several issues in this paper that need resolving before it can be published.
First, the “English” is in need of repair – later, I will show a few examples in particular lines.
Second, there is considerable discussion of the “models” used for comparison to the measurements. The one model, MODTRAN, is well-documented in the literature as to its performance and the details of the model are well-documented in the description that accompanies the model distribution. The second “model” is alluded to but never actually defined – it is only described as a Mie model. So, the question is whether the authors have written the code for a model that is based on Mie theory or whether they use someone else’s code. And, I am not intimate with MODTRAN; however, I would imagine that it is possible to use MODTRAN in an only Mie “mode” – perhaps, not. Anyway, I believe that this needs clarification. And, if MODTRAN can be run in Mie only mode, was that configuration compared to the other “Mie Model”?
Another issue concerns the use of the term “granulometer” which does not seem to have wide use in the literature with which I am familiar. It seems as though what is being used is a “particle size analyzer” (which may be abbreviated PSA) which is in more widespread use. In fact, the purported manufacturer of the “granulometer” (PALAS) does not use the term on its web site (a search there, yields a null response).
Several equations are missing a reference to their source…
Now some more specific comments relating mostly to the use of “English”:
Line 7 – are you only proposing this study; or, are you reporting on its results?
Line 13 – sentence is not clear as to meaning; for example, “fog criterion” is undefined and the use of the term “no significant difference” is usually used to describe the results of a statistical test comparing two different results – not sure what it means here or later in the paper (line 439)
line 26 – the word “propose” is the wrong choice – should be something like “…we examined…”
Line 32 – “the problem” should be spelled out here – restate.
Line 34 – remove word “then”
Line 35 – remove word “already”
Line 45 – I do not understand what is being said in this sentence…
Line 51, 58, 67 – paragraph indents missing…
Line 83 – “compare” should be “compares”
Line 88 – “propose” should be something like “achieve”….
Equation 1 – you need a reference for this equation…
Equation 2 – you need a reference here…
Figure 2 – cannot see the “yellow”…
Equation 3 – needs reference…
Between line 178 and line 179, just after “[9]”, tense is wrong – should be “was” instead of “will be”…
Line 195 – “possible” should be “made possible”….
Line 258 – “sens” should be “sense”…
Line 266 – “spectums” should be “spectra”…Also, the sentence that begins on this line needs a reference validating the claim of “unprecedented” which, in any case, this sentence is not really needed.
Line 281 – I do not understand what is meant by “two dissipations”…
Line 288 – this sentence does not make sense to me…
Line 293 – phrase at end of this sentence does not make sense…
Line 313 – the sentence that begins on this line is incorrect; in no way does a correlation coefficient show “cause”!...
Line 321 – A reference is needed that describes or justifies how an R less than 0.9 suggests a poor relationship and, if data with an R < 0.9 is deemed uncertain or unusable, then why continue to show it?
Line 392 – instead of “following”, I believe that the authors mean “preceeding”
Line 395-397 – this sentence needs to be rewritten – not sure what is meant by “first”…
Line 398 – Is the “unexpected local behavior” that due to water vapor? If so, then I hardly think it “unexpected”…
Line 408 – the beginning of the sentence here has issues with grammar…
Line 420 – this paragraph needs to be rewritten as it is “rambling”…
Line 436 – first sentence is unnecessary and may be removed…
Line439 – Again, the use of the phrase “no significant difference” is, I believe, reserved for comparisons made in statistics…
Line 463 – the asterisk just after Microns is, I believe, out of place…
Line 493 – this reference [16] is not used in the text; perhaps, it should be…
Author Response
The authors would like to thank reviewers again for the second proofreading. The new comments that have been made are always in the right direction, and have greatly improved the article. Majors modifications are :
the term “granulometer” has been changed for “particle size analyser (PSA)” in the whole paper.
“English” improvement.
Precisions about the Mie Model.
Specific responses:
Ok, the “English” was corrected following your comments.
Second, there is considerable discussion of the “models” used for comparison to the measurements. The one model, MODTRAN, is well-documented in the literature as to its performance and the details of the model are well-documented in the description that accompanies the model distribution. The second “model” is alluded to but never actually defined – it is only described as a Mie model. So, the question is whether the authors have written the code for a model that is based on Mie theory or whether they use someone else’s code. And, I am not intimate with MODTRAN; however, I would imagine that it is possible to use MODTRAN in an only Mie “mode” – perhaps, not. Anyway, I believe that this needs clarification. And, if MODTRAN can be run in Mie only mode, was that configuration compared to the other “Mie Model”?
“Equations 3 to 9 were implemented in C++ to determine the extinction coefficient for each wavelength from the DSD measurements. We then refer to the Mie model in the following.” Added on line 179
“It is important to note that when we omit the second term of the right-hand side of the equation 10 it goes back to our Mie model.” Added on line 191
We not use a “partial Mie model” on MODTRAN because we wanted to know exactly what was implemented. Then we write our own code in C++ to implement the “Mie model”.
Another issue concerns the use of the term “granulometer” : ok, your comment is totally justified, we change it for particle size analyzer (PSA).
Several equations are missing a reference to their source : ok sources were added.
Now some more specific comments relating mostly to the use of “English”:
Line 7 – Ok, new formulation :
“The influence of wavelength on light transmission in fog is then studied and results reported.”
Line 13 and 439 – Ok, we modify the formulation :
“Finally, the wavelength choice for lidars should not be made taking into account the fog conditions: there is a small difference (< 10%) between the extinction coefficients at 905 nm and 1550 nm for the same emitted power.”
line 26 – Ok, thanks.
Line 32 – Ok :
“Scientific studies concerning the extinction coefficient of light in fog have already been carried out, without fully addressing the problem: either because the wavelength ranges are limited, or because the DSD is not taken into account, or because the comparison between models and experimental measurements is not addressed.”
Line 34 – ok
Line 35 – ok
Line 45 – ok
Line 51, 58, 67 – ok
Line 83 – ok
Line 88 – ok thanks.
Equation 1 – ok
Equation 2 – ok
Figure 2 – ok, the colour has been changed.
Equation 3 – ok
Between line 178 and 179: ok
Line 195 – ok
Line 258 – ok
Line 266 – “spectums” should be “spectra” : ok
Line 266 - Also, the sentence that begins […] this sentence is not really needed. Ok, we removed it.
Line 281 – ok
Line 288 – ok
Line 293 – ok
Line 313 – ok
Line 321 – A reference is needed that describes or justifies how an R less than 0.9 suggests a poor relationship and, if data with an R < 0.9 is deemed uncertain or unusable, then why continue to show it?
Ok we modify it:
“When the correlation coefficient is lower than 0.9 (R≤0.9), the measurements are labelled as uncertain data. The value of 0.9 is arbitrarily chosen but allows a better reading of the figure 9.”
Line 392 – ok
Line 395-397 – ok
Line 398 – ok
Line 408 – ok
Line 420 – this paragraph needs to be rewritten as it is “rambling”…
ok :
“Two main questions are still pending and will deserve to be addressed in further research. First, it seems important to compare the advection and radiation fogs produced on the PAVIN BP platform with natural fogs. This study showed the importance of the DSD on the extinction coefficient for near infrared wavelengths. It is then necessary to measure the DSD of natural fogs and compare it to the one of fogs produced in the PAVIN BP platform. Secondly, the good correlation between the measurements of the coefficient correlation made with a spectroradiometer and the combination model + PSA leads us to think about posing the opposite problem. It would then be possible to use the spectroradiometer as a PSA, by optimizing a theoretical DSD to obtain the measured extinction coefficient as a function of wavelength.”
Line 436 – ok
Line439 – ok
Line 463 – ok
Line 493 – reference 16 is cited for equations (4) (5)
Round 3
Reviewer 3 Report
Comments concerning third review of 484502:
This manuscript (#484502) is an attempt to determine the extinction coefficient of light in fog as a function of wavelength. There are several issues in this paper that need resolving before it can be published as was indicated in the first and second review. A major issue is still the use of “English” as many of the problems seen earlier still exist. I will attempt to delineate some of these and other issues in my comments below.
Line 28 – This sentence needs to be rewritten so as to be clear about the MOR and fog density, etc.
Line 37 to line 45 – it is unclear as to whether the discussion relates to reference [2] because of the “English” and sentence structure…
Line 53 – “the work” should probably be “that work”…
Line 55 – “this study” should probably be “that study”…
Line 57 – the phrase “the advanced results” should probably be “…the results presented.”
Line 82 – a more proper word for “rigorous” might be “complete; not sure that I would call the author’s approach “rigorous”…
Line 104 – “throughout” should probably be “through” and I would guess that this equation is not just for fog…also, if you go to Line 489 – this reference ([10]) is incorrect as it leads nowhere; after about 1-1/2 hours on the internet, I found what might be a more current version of the one that the authors attempted to reference: https://library.wmo.int/index.php?lvl=notice_display&id=12407 – searching Google Scholar, I did find about 4 similar references; probably, all copied from one another but not leading to an actual document…
Line 136 – 145 – the authors imply that attenuation is only caused by scattering whereas absorption also leads to attenuation of the signal…
Line 150 – “multi” should probably be “multiple”…
Line 167 – is the “m” being used as a both a variable and a measure of distance? (for example is m equal to “meter” in some places and as an index elsewhere?) if so, needs to be fixed…
Line 172 – sentence does not make sense…
Line175 – this paragraph does not make sense…
Line 202 – the sentence that begins on this line does not make sense…
Line 208 – “tape” should be “tap”,,, and the sentence that begins on this line needs to have the “English” fixed…
Line 250 – “The Figure 7 present…” should probably be “Figure 7 presents…”
Line 251-2 – awkward “English”
Line 254 –“in meteorological” should be “in the meteorological” and “…explains that…” should be “…explains the reason that…”
Line 270 – what is the meaning of “…the data in an original way.”?
Line 281 – “more or less” should be “plus or minus”
Figure 7 -- caption is incomplete…
Table 1 –in the heading, instead of “V(m)” using “MOR” or “MOR, V(m)” or similar would be more instructive…
Line 300 – the regression details are not given and, it may not be critical to this paper; but, if the authors used the ordinary least squares regression then they have violated one of its basic assumptions that there are not errors in both variables…
Line 312 – sentence does not make sense
Line 314 – the correlation coefficient does not measure degree of causality!
Table 2 – I believe that the correlation coefficient is not intimately related to the regression and its value would be independent of the regression performed as long as the data have a linear relationship…also, there is not discussion of the low values of 0.77, 0.64, 0.66, etc
Line 326 – this whole paragraph seems out of place especially relative to the discussion in the next paragraph (line 343)
Line 337 – 340 – I do not understand what you are doing here…
Line 345 – not a new paragraph – part of preceding paragraph…and, if I understand what the authors are saying from line 345 to 349 then I disagree with what they say as I “read” the plots in figure 9…
Line 367-8 – where does this come from; who says so…?
Figure 9 – the variable ordinate axis scale from plot to plot makes it difficult to compare the results…the legend in this figure also still has the term “Granulometer”… and the caption is incomplete…
Line 372 – “model” should be “models”…
Line 379 – Why?
Line 382 – “confirm” should be “confirms”…
Line 389 -390 – based on your statement, I infer that the fog produced in the PAVIN BP is different from real fog…is this true?
Line 394-395 – paragraph is unnecessary….
Line 397 – this whole paragraph seems off the mark…looking at Figure 9, it would seem as though the two models agree quite well with one another by differ (both in the same way and amount) in the results for the two fogs (which probably is expected because of the particle size…however, they both disagree with the direct measurement…
Line 424 – sentence is awkward and begs the question of what is different between the laboratory fog and natural fog.
Line 440 – not sure that I would agree…
Line 500 – in this reference ([16]), it appears as though the correct report number is “2002-11” – not sure to what the number “8” at the end of the reference means…Also, for such an obscure publication, it would probably help to provide a more complete source information…based on the authors reference, I found on-line at the University of Arizona (USA)…
Author Response
We thank you for this new proofreading round. We have tried to take into account all your comments. We also transmitted the article to our translation service for a complete correction (by a native English speaker), many English mistakes where then corrected. You will find our comments on the most critical points below.
Line 167 – variable « m » is now « gamma »
Table 1 – ok, we have chosen “MOR, V(m)”
Line 300 – the regression details are not given and, it may not be critical to this paper; but, if the authors used the ordinary least squares regression then they have violated one of its basic assumptions that there are not errors in both variables…
Sorry, we do not understand your remark. In the article, we used an ordinary least squares regression. We do not see how the linear regression we are doing is not justified, because our model is a linear model (see Eq 14).
Table 2 – I believe that the correlation coefficient is not intimately related to the regression and its value would be independent of the regression performed as long as the data have a linear relationship…
We used a linear regression, as the theoretical model presented is well linear (see Eq 14). We have also specified the term linear regression in the article.
also, there is not discussion of the low values of 0.77, 0.64, 0.66, etc
From lines 214 to 316, we gave an explanation for low R values when the MOR is high.
Line 337 – 340 – I do not understand what you are doing here.
As it is done in the literature (e.g. MODTRAN), the extinction is normalized with respect to a reference value, here given by the transmissiometer. This value is therefore measured at 550nm, as specified by the manufacturer.
Line 345 – not a new paragraph – part of preceding paragraph…and, if I understand what the authors are saying from line 345 to 349 then I disagree with what they say as I “read” the plots in figure 9…
The wavelength variations for each type of fog (advection and radiation) are the same between model+PSA measurements and spectoradiometer measurements. However, the amplitudes of these variations remain different between the measurements from the model+PSA and tha spectoradiometer.
Figure 9 – the variable ordinate axis scale from plot to plot makes it difficult to compare the results…the legend in this figure also still has the term “Granulometer”…
There is a factor 10 on the y-axis scale between sub-figures MOR=15m and MOR=175m. It is therefore not possible to fix the scale for all sub-figures. To compensate, we choose to keep a constant step for the sub-grid that allows a better understanding of the differences from one sub-figure to another.
Line 389 -390 – based on your statement, I infer that the fog produced in the PAVIN BP is different from real fog…is this true?
The fog produced in the PAVIN BP facility is as similar as possible as real fog. However, some research work remains to compare fogs (natural ones and artificial ones) and better validate the similarity. Hence, the perspective proposed at the end of the article.
Line 397 – this whole paragraph seems off the mark…looking at Figure 9, it would seem as though the two models agree quite well with one another by differ (both in the same way and amount) in the results for the two fogs (which probably is expected because of the particle size…however, they both disagree with the direct measurement…
As explained in the body of the article (line 343), we believe that the variations are similar. We agree that the amplitude is different, but in our opinion, what we call the "trend" or "behaviour" is the same.
Round 4
Reviewer 3 Report
Comments concerning re-review of manuscript 484502:
The title of this paper is: Light transmission in fog - influence of wavelength on the extinction coefficient. My copy, v4, does not have line numbers; so, it will be somewhat difficult to make comments concerning specific content. I would like to thank the authors for all of their extra work on this paper, I really appreciate it.
My first comment addresses the authors not quite understanding a comment that I made earlier regarding the use of linear regression. As I said earlier, for this particular work, my comment may not result in a significant difference. Anyway, In those cases where there is uncertainty (or, errors) in both variables it is more appropriate to use an orthogonal regression technique. In fact, because there are errors in both variables, the results an ordinary least squares (OLS) regression analysis are invalid. Further discussion can be found in:
Isobe, T., Feigelson, E. D., Akritas, M. G. & Babu, G. J. (1990). Linear regression in astronomy. I. The Astrophysical Journal, 364, 104–113
Feigelson, E.D., and G.J. Babu, (1992). Linear regression in astronomy, II, The Astrophysical Journal, 397, 55-67.
And
Wu, C. and Yu, J. Z.: Evaluation of linear regression techniques for atmospheric applications: the importance of appropriate weighting, Atmos. Meas. Tech., 11, 1233-1250, https://doi.org/10.5194/amt-11-1233-2018, 2018.
Cantrell, C. A.: Technical Note: Review of methods for linear least-squares fitting of data and application to atmospheric chemistry problems, Atmos. Chem. Phys., 8, 5477-5487, https://doi.org/10.5194/acp-8-5477-2008, 2008.
Keleş, T. (2018). Comparison of Classical Least Squares and Orthogonal Regression in Measurement Error Models, International Online Journal of Educational Sciences, 10(3), 200-214.
Gillard, J.: An Historical Overview of Linear Regression with Errors in both
variables. Cardiff University, School of Mathematics, TR (2006):
So, some specific comments:
1 – at end of abstract, I think that you should add the phrase “in fog” so that it reads” “…emitted power in fog.”
2 – first paragraph on page 2: at end of paragraph, it seems as though the phrase “…, fogs with the same MOR…” should have at the end of that phrase: “for different wavelengths.”
3 – in the 4th paragraph, the variable “V” is introduced without prior definition, I believe.
4 – in figure 3, the legends for each graph should have definitions (droplet diameter, wavelength)
5 – at the bottom of page 6, the first mention of Figure 5 is made – the figure is too far away from the text and the reference to figure 5 comes before figure 4…
6 – in section 3.1, first paragraph, the phrase “…by increments…” is used – need to specify the increments…
7 – If I am reading Figure 9 correctly, in the third paragraph on page 13, you talk about the two methods agreeing; however, I see large differences between modeled and measured…
8 – in Figure 9 caption, you say that the “…grid is with constant steps of 0.01m -1” yet, panel (a) had grid steps of 0.03 and panel (b) has grid steps of 0.05…Also, in the label for the ext. coeff., the “-1” is not superscripted…)
9 – in first paragraph on page 15, the sentence “However, the MODTRAN model tends to globally underestimate …compared to the Mie model.” If this were true, then as I understand it, the solid line in Figure 9 would always be less that then dashed line and it seems to me as both lines mostly overlay on each other…
10 – on page 16, second full paragraph, “…problem the over way around.” should be “…problem the other way around.”
Author Response
Thank you again for all your precise and helpfull comments.
Thank you for all references about orthogonal regression. Then, we change from ordinary least square regression to orthogonal regression. It changes almost nothing to the the curves (only some points move a little).
1 – ok, you’re right it’s more clear with « in fog ».
2 – idem, ok.
3 – ok. « meteorological optical range (MOR), denoted by V, from 15m to 175m; » We added «denoted by V» at first apparition of MOR.
4 – ok, done.
5 – ok Figure 4 and Figure 5 are moved and inverted.
6 – ok we add it.
7 – Yes there is a large difference, but the trend is the same, it is what we say in the article.
8 – we correct it.
9 – modification : « the MODTRAN model tends to globally slightly underestimate the extinction coefficient compared to the Mie model, except for the molecular scattering peaks. »
10 – on page 16, second full paragraph, “…problem the over way around.” should be “…problem the other way around.”